

# Environmental conditions of a salt-marsh biodiversity experiment on the island of Spiekeroog (Germany)

Oliver Zielinski[1], Daniela Meier[1], Kertu Lõhmus[2], Thorsten Balke[3], Michael Kleyer[2], Helmut Hillebrand[1,4]

[1]Institute for Chemistry and Biology of the Marine Environment, University of Oldenburg, Wilhelmshaven, 26382, Germany
[2]Institute of Biology and Environmental Sciences, University of Oldenburg, Oldenburg, 26129, Germany
[3]School of Geographical and Earth Sciences, University of Glasgow, Glasgow, G128QQ, UK
[4]Helmholtz-Institute for Functional Marine Biodiversity, University of Oldenburg, Oldenburg, 26129, Germany

*Correspondence to*: Oliver Zielinski (oliver.zielinski@uol.de)

**Abstract.** Field experiments investigating biodiversity and ecosystem functioning require observation of abiotic parameters, especially when carried out in the intertidal zone. An experiment for biodiversity-ecosystem functioning at the intersection of land and sea was set up in the intertidal zone of the back-barrier salt marsh of Spiekeroog Island in the German Bight. Here we report the accompanying instrumentation, maintenance, data acquisition, data handling and data quality control as well as monitoring results observed over a continuous period from September 2014 through April 2017. Time series of abiotic conditions were measured at several sites in the vicinity of newly built experimental 'salt-marsh islands' on the tidal flat. Meteorological measurements were conducted from a weather station (WS, doi: 10.1594/PANGAEA.870988), oceanographic conditions were sampled through a bottom mounted recording current meter (RCM, doi: 10.1594/PANGAEA.877265) and a bottom mounted tide and wave recorder (TWR, doi: 10.1594/PANGAEA.877258). Tide data are essential to calculate flooding duration and flooding frequency with respect to different salt marsh elevation zones. Data loggers (DL) for measuring water level (DL-W, doi: 10.1594/PANGAEA.877267), temperature (DL-T, doi: 10.1594/PANGAEA.877257), light intensity (DL-L, doi: 10.1594/PANGAEA.877256) and conductivity (DL-C, doi: 10.1594/PANGAEA.877266) were deployed at different elevational zones within the experimental islands and the investigated salt marsh plots.

## 1 Introduction

Biodiversity is changing at an unprecedented high rate (Mace et al. 2005) reflecting the anthropogenic alteration of Earth's ecosystems (Vitousek et al. 1997). In consequence, research on biodiversity - ecosystem function (BEF) relationships has become a major facet of ecology and evolutionary biology (Balvanera et al. 2006; Cardinale et al. 2006; Hillebrand & Matthiessen 2009, Cadotte et al. 2008; Gravel et al. 2011a). The research on BEF has been dominated by experimental studies manipulating the number of species in a trophic group (Hillebrand & Matthiessen 2009). However, the change in the number of species is not the predominant pattern in biodiversity change, which more frequently comprises altered dominances and species turnover (Hillebrand et al. 2017). Moreover, the mechanisms involved in altering species composition, i.e. immigration



and extinction, are usually experimentally prohibited in most BEF studies, e.g. via weeding. On the other hand, there have been recent advances in understanding the interaction between regional community assembly, local dynamics and the biodiversity-functioning aspects (Hodapp et al. 2016, Leibold et al. 2017) which should be incorporated in BEF experiments. The intertidal zone represents an interface between terrestrial and marine processes and biodiversity. This area is sensitive to

climate change and is heavily impacted by anthropogenic activities (Cooley and Doney 2009; Ekstrom et al. 2015; Haigh et al. 2015; Mathis et al. 2015). Within the near-shore environment, salt marshes have increasingly gained attention in times of sea level rise but studies on metacommunity dynamics remain scarce.

The aim of the project BEFmate „Biodiversity - ecosystem functioning across marine and terrestrial ecosystems" was to quantify the dynamics of biodiversity and associated functions of salt marsh and tidal flat ecosystems. For this purpose, a series

of 12 artificial islands (I 1-12) constructed from galvanized steel and filled with sediment from the tidal flat were set up in September 2014 on the back-barrier tidal flat of Spiekeroog island in the German East Frisian Wadden Sea (Balke et al. 2017). The experimental salt marsh islands were mirrored by 12 salt marsh enclosed plots (S 1-12) at three different elevations located within the nearby salt marsh on Spiekeroog. Here we report abiotic parameters observed from 23 sensors installed either near the experimental islands, within the island structures themselves or within the nearby salt marsh as well as meteorological data

from a locally installed weather station. We describe the instrumentation, data handling and results observed over a period of 32 months starting from mid of September 2014. We will further discuss the data and results with respect to validity and potential limitations of the observational setup.

## 2 Materials and Methods

### 2.1 Research Area

The island of Spiekeroog is located in the Southern North Sea (Fig. 1) and is part of the Wadden Sea that has been renowned as an UNESCO world natural heritage since 2012. Twelve experimental islands (I1 - I12) were built in the back-barrier tidal flats at distances of 240 m (I12) to 460 m (I1) from the southern salt marsh of the island of Spiekeroog. The experimental islands were distributed unevenly over 810 m from East to west at an elevation of 0.8 m NHN, with a mean tidal range of 2.7 m (Fig. 2, numbered 1-12 from East to West). The islands were built in a northeast-southwest direction with the lowest

elevation at the northeast end of the island. The actual sensor position on the islands was determined by the local bathymetry since the experimental islands encompass three different elevation levels (Fig. 3), reflecting pioneer zone (Pio; 1.5 m NHN), lower salt marsh zone (Low; 1.8 m NHN) and upper salt marsh zone (Upp; 2.1 m NHN) of natural salt marshes. Half of the experimental islands were filled with mudflat sediment and left bare, whereas half of them were additionally transplanted with sods from the lower salt marsh zone of the natural adjacent salt marsh. These islands represent a treatment of dispersal

limitation, constraining community assembly on the islands. Additionally 12 equally treated salt marsh enclosed plots (S 1-12) were created (Fig. 2) that reflect unlimited dispersal. In addition to experimental plots, six control plots (C) per each salt marsh elevation zone were marked but left natural to compare established communities with the community assembly on



dispersal limited island and unlimited saltmarsh plots. The experimental design and setup of the artificial islands are not subject of this work and are described in detail in Balke et al. (2017). Abiotic conditions were measured at several sites due to the involvement of wide selection of parameters. Details concerning the individual sensors, their location, data provided and the associated methods of data handling are provided in the following subsections. All positions, coordinates as well as elevations

of sensors are indicated and provided in table 1.

## 2.2 Instrumentation and Data Processing

### 2.2.1 Weather Station (WS)

Meteorological data were collected near the experimental setup (see table 1) with a locally installed weather station located approximately 500 m north of the southern shoreline (53°45'57.10" N, 007°43'34.11" E). The system was installed at the end

of a glass fibre pole at a height of 10 m. The weather station system used here was a ClimaSensor US 4.920x.00.00x, that was pre-calibrated by the manufacturer (Adolf Thies GmbH & Co. KG, D-Göttingen). Data were recorded and saved within the Meteo-Online (V4.5.0.20253) software in a sampling interval of 1 minute with an averaging time of 10 seconds, with date and time were given in UTC. Position, solar azimuth angle and solar elevation were derived from the internal GPS-system. Ultrasonic propagation time measurements were used for the determination of wind speed and direction. Two sensors were

integrated for measuring air temperature and relative humidity (precision combination sensor) as well as atmospheric pressure (micro-electro-mechanical system (MEMs) technology). For recording and calculating precipitation the back reflected signal of a Doppler radar was used. Additional four photo sensors were used for the identification of light direction and light intensity. Post-processing of collected data was done using MATLAB (R2012b). Further quality control was performed by a) erasing negative readings and data covering maintenance activities, b) visual inspection of the overall dataset and c) removal of

outliers, defined as data exhibiting changes of more than two standard deviations within one time step. As the weather station was not oriented directly towards North, a manually North correction had to be done afterwards for accurate wind direction values (+ 20° from Nov. 2014 to Mar. 2016 and + 10.1° since Oct. 2016).

### 2.2.2 Recording Current Meter (RCM)

A RCM9 LW recording current meter (AADI, Aanderaa, Bergen/Norway; RCM DCS 4220) with additional temperature

(3621), conductivity (3919), and pressure (4017) probes was deployed for deriving hydrographic conditions (see table 1). The device was bottom mounted through a buried H-anchor between islands 6 (I6) and 7 (I7) (53°45'29.34" N, 007°43'16.50" E), approximately 35 m southeast of island 7 and 50 m southwest of island 6 in a shallow tidal creek (0.71 m NHN). The position was derived from a portable DGPS-system (Leica Differential-GPS, SR530). The acoustic sensor head was placed 0.4 m above the sediment at 1.2 m above mean low water height. In consequence, the sensor head fell dry during low tide and data had to

be examined and eliminated accordingly. The sensor was pre-calibrated by the manufacturer. Recorded data were internally logged on a memory card (DSU 2990 E), with a sampling interval of 10 minutes, until readout with the Data Reading Program



DRP 5059 software. Date and time was given in UTC. Post-processing was performed using MATLAB (R2012b) to remove low tide data. Conductivity values were used as an indicator of low tide periods. Data were erased when conductivity values fell below 25 mS/cm. Further quality control was applied as described in section 2.2.1 for the weather station (WS).

### 2.2.3 Tide and Wave Recorder (TWR)

Local tide and wave conditions were recorded with a RBRduo TD | wave sensor (RBR Ltd., Ontario/Canada). The sensor was bottom mounted in a shallow tidal creek (0.71 m NHN) through a steel girder (buried 0.3 m deep in the sediment) next to the RCM 9 LW recording current meter between island 6 and 7 (53°45'29.34"N, 007°43'16.50"E) (see table 1). The sensor was thus positioned 10 cm above sediment surface, as was determined by using a portable differential GPS (Leica Differential-GPS, SR530). Alike the recording current meter, this resulted in falling dry during low tide and data had to be flagged

accordingly. The sensor was pre-calibrated by the manufacturer and the sampling rate was 3 Hz with 1024 samples per burst at a sample interval of 10 minutes. Recorded data were internally logged until readout with the Ruskin (V1.13.10) software and post-processed using MATLAB (R2012b). Date and time was given in UTC. For accurate depth calculations raw pressure data were manually corrected for atmospheric pressure derived from the locally installed weather station. Low tide data was not removed, but was easily identified through the manually calculated water depth data, where all depths < 0.05 m represented

low tide data. Again, quality control was applied as described in section 2.2.1 for the weather station (WS).

### 2.2.4 Data Logger (DL)

Several data loggers were installed within the experimental islands as well as the salt marsh enclosed plots for the observation of groundwater level, temperature, light and salinity (see table 1). In all cases date and time is given in UTC and all post-processing was performed using MATLAB (R2012b). Quality control was applied as described in section 2.2 for the weather

station (WS).

To get a continuous observation of flooding and the groundwater levels inside the experimental islands as well as in the salt marsh, pressure loggers were deployed in dip wells within the experimental setup at different elevational levels. Six HOBO® U20L Water Level Logger (onset® HOBO® Data Loggers, Bourne, MA/USA; S/N 10685287, 10685288, 10685289, 10685290, 10685291, 10685292; Hobo-P) as well as a DEFI-D Miniature Pressure Recorder (S/N OA5K008; DEFI-D) were

deployed. All water level loggers were pre-calibrated by the manufacturer. Recorded data were internal logged until readout afield with the Hobo Underwater Shuttle (U-DTW-1) and the HOBOware Pro (V3.7.4) software respectively with the DEFI Series software (V1.02). For depth calculations, pressure data were manually corrected by atmospheric pressure. Accordingly, one of the HOBO® U20L Water Level Logger was installed outside the dip wells at a higher elevation, attached on a steel pole at the upper zone of island 3. All loggers were initially calibrated to get the exact height inside the dip well. Data of one

Hobo Logger had to be corrected with -0.082 m due to a wrong initial measurement. Further corrections were applied to the DEFI-D logger since a manually correction with atmospheric pressure was not possible. As the local maxima of the other



water level logger were very similar, a mean value of the other loggers was calculated to correct the DEFI-D logger values with +0.25 m.

Temperature in sediment surface layer (in approximately 0.05 m depth) was measured with six DEFI-T Miniature Temperature Recorders (JFE Advantech Co., Ltd., Tokyo; DEFI-T). The manufacturer pre-calibrated temperature recorders were installed

within the experimental island and salt marsh enclosed plots at different elevation levels. Recorded data were internally logged until readout with the DEFI Series software (V1.02).

Light availability was measured with six locally installed light intensity loggers within the experimental islands as well as in the saltmarsh plots at different elevation levels. The DEFI-L Miniature Light Intensity Recorder (JFE Advantech Co., Ltd., Tokyo; DEFI-L) used here were pre-calibrated by the manufacturer. Recorded data were internal logged until readout with the

DEFI Series software (V1.02). Due to different calibrations for under water or above water application, raw data were processed differently and merged depending on water levels. The processed pressure and depth data of the RBRduo TD | wave sensor was used to identify flooding times and durations of the different elevations.

Two HOBO conductivity loggers (Onset Computer Corporation, Bourne, MA/USA) were installed inside of dip wells within the experimental islands as well as in the salt marsh enclosed plots at the pioneer zone. Conductivity logger used here were

Hobo U24 Conductivity Logger U24-002-C (S/N 10570000, 10599255). The conductivity loggers were pre-calibrated by the manufacturer. Recorded data were internal logged until readout afield with the Hobo Underwater Shuttle (U-DTW-1) and in the following with the HOBOware Pro (V3.7.4) software. An automatic calculation of salinity was conducted within the software according to PSS-78 using the measured conductivity and temperature. Due to fluctuations in ground water level, conductivity loggers periodically fell dry, especially in the beginning of the deployment. Data until October 2015 are therefore

very scattered. Thereupon the depth of conductivity loggers were adjusted to the bottom of the dip wells assuring a constant coverage with water. Data from dry sensors was removed, using a salinity of 20 psu as a threshold value. As a reference, soil samples of all plots within the experimental islands and salt marsh enclosed plots were sampled to analyze pore water salinity in laboratory (data not shown here). Comparative data of meteorological and hydrographic conditions for validation processes were taken from the nearby Time Series Station - Spiekeroog (TSS) at Otzumer Balje (Holinde et al. 2015, Baschek et al.

2017).

## 2.3 Data Provenance, Structure and Availability

All datasets described herein are available at the World Data Center PANGAEA (www.pangaea.de) searching for "BEFmate" as identifier (Zielinski et al. 2017a – 2017g). Due to extensive datasets as well as individual questions of potential users a collection (parent) for each sensor (type) was created. Each parent serves as a basis where data are divided into monthly

sections enabling an easy selection of required time periods.

### 2.3.1 Weather Station (WS)

Data of meteorological observations are available at PANGAEA for Nov 2014 to Apr 2017 (doi: 10.1594/PANGAEA.870988). Gaps in the dataset resulted from a malfunction of the pressure sensor and an adjacent maintenance and re-calibration by the manufacturer. The structure of dataset is shown in table 2.

### 2.3.2 Recording Current Meter (RCM)

Continuous current data are available on PANGAEA for Sep 2014 to Oct 2015 (doi: 10.1594/PANGAEA.877265). There are additional datasets before the main sampling period for Nov/Dec 2013, Mar 2014, and Jun 2014. These data were used as a basis for deciding on the final placement and orientation of the experimental island. Gaps in the dataset are resulting of local readouts and maintenance. Low-tide data were removed from the dataset. Data after October 2015 is missing due to sensor malfunctions. Structure of the datasets is shown in table 3.

### 2.3.3 Tide and Wave Recorder (TWR)

Tide and wave data are available on PANGAEA for Oct 2014 to Apr 2017 (doi: 10.1594/PANGAEA.877258). Gaps in the dataset are resulting from local readouts and maintenance. Low-tide data was not removed from the dataset. The depth was calculated from raw pressure, which was before corrected with atmospheric pressure. Structure of the datasets is described in table 4.

### 2.3.4 Water Level Loggers (DL-W)

Data for groundwater level within the experimental islands and salt marsh enclosed plots are available on PANGAEA for Jun 2015 to Apr 2017 (doi: 10.1594/PANGAEA.877267). Gaps in the dataset are resulting of local readouts and maintenance. For depth calculations, pressure data were manually corrected by atmospheric pressure. Each dataset includes date/time and water level (water level [m]) (table 5).

### 2.3.5 Temperature Loggers (DL-T)

Temperature data for the surface sediment layer of the experimental islands and salt marsh enclosed plots are available on PANGAEA for Sep 2014 to Apr 2017 (doi: 10.1594/PANGAEA.877257). Gaps in the dataset are resulting from local readouts and maintenance. Each dataset includes date/time, depth in sediment (Depth [m]) and temperature in sediment (t [degC]) (table 6).

### 2.3.6 Light Loggers (DL-L)

Measured light availability within the experimental islands and salt marsh enclosed plots are available on PANGAEA for Sep 2014 to Apr 2017 (doi: 10.1594/PANGAEA.877256). Gaps in the dataset are resulting from local readouts and maintenance.

Due to different calibrations for under water or above water application, raw data were processed differently and were merged depending on low or high water times. Each dataset includes date/time and light intensity (Io [µmol/m²s]) (table 7).

### 2.3.7 Conductivity Loggers (DL-C)

Data of conductivity measurements inside dip wells within the experimental islands and salt marsh enclosed plots are available
on PANGAEA for May 2015 to Apr 2017 (doi: 10.1594/PANGAEA.877266). Gaps in the dataset are resulting of local readouts and maintenance. Low-tide data was removed from the dataset. An automatic calculation of salinity were performed within the HOBOware Pro (V3.7.4) software according to PSS-78. Each dataset includes date/time, depth in sediment (Depth [m]) and Salinity (Sal [-]) (table 8).

### 3 Data availability

Sensor operation encompasses the time span of 32 month from 18th Sep 2014 until 18th Apr 2017 (944 days). Figure 4 illustrates the data availability over the whole period and table 9 provides total availability in days as well as in percent.

### 3.1 Weather Station (WS)

Within this time frame meteorological data was available from 19th Nov 2014 to 18th Apr 2017 on 571 days (corresponding to 60.49 % availability). Days absent resulted from malfunctions and maintenances from 29th Jan 2015 to 11th Mar 2015, 04th
Nov 2015 to 14th Nov 2015, 25th Mar 2016 to 28th Oct 2016, 01st Feb 2017 to 21st Mar 2017.

### 3.2 Recording Current Meter (RCM)

Current meter operation was possible from 18th Sep 2014 until 06th Oct 2015 on 335 days (35.49 %). Further operations occurred in Nov/Dec 2013, Mar 2014 and Jun 2014 for preliminary investigations. Missing days were resulted from local maintenance and readouts. Due to a malfunction, the sensor was not operating since Oct 2015.

### 3.3 Tide and Wave Recorder (TWR)

Continuous observations of wave and tide data were feasible from 01st Oct 2014 to 18th Apr 2017 on 812 days (corresponding to 86.02 % availability). Days absent were a result of local maintenance and readouts as well as some extended services from 19th Oct 2014 to 28th Oct 2014, 17th Aug 2015 to 23rd Sep 2015, and 08th Nov 2016 to 24th Jan 2017.



### 3.4 Data Logger (DL)

#### 3.4.1 Water Level Loggers (DL-W)

Within the total time period water level measurements were performed from 20th Jun 2015 to 18th Apr 2017 on 578 days (572 days within the salt marsh enclosed plot) resulting in a 61.23 % (60.59 %) availability. Days absent resulted from malfunctions

and maintenances from 16th Feb 2016 to 18th May 2016 as well as some local readouts. Groundwater level data of the DEFI-D logger ends at 10th Jan 2017 due to a missing readout in April 2017. The coverage of water level measurements is therefore 50.85 % (480 days).

#### 3.4.2 Temperature Loggers (DL-T)

Recording of surface layer temperature were available on 931 days, excluding the period from 10th Jan 2017 to 24th Jan 2017

due to maintenance (98.62 % availability). However, temperature data of two loggers within the salt marsh enclosed plots (S2Upp and S2Low) are exhibiting a shorter time of deployment. An application was possible at 295/380 days (31.25 % / 40.25 % availability) from 16th Dec 2014 to 06th Oct 2015 (S2Upp and S2Low) and further from 24th Jan 2017 to 18th Apr 2017 (S2Low).

#### 3.4.3 Light Loggers (DL-L)

Light intensity and availability was recorded over the whole period from 18th Sep 2014 to 18th Apr 2017 for 931 days, except from 10th Jan 2017 to 24th Jan 2017, due to maintenance (98.62 % availability). Two loggers (I3 Pole, Seafloor) were first installed on 7th Oct 2015 with a total application time of 547 days corresponding to 57.94 % availability. One light logger within the salt marsh enclosed plot (S3Low) had to be removed from 19th Nov 2014 to 16th Dec 2014 due to maintenance. One light logger within the experimental island (I3Low) had to be temporarily removed between 23rd Sep 2015 and 11th Apr

2016.

#### 3.4.4 Conductivity Loggers (DL-C)

Data of the conductivity logger within the experimental island (I3Pio) is available from 06th May 2015 to 16th Feb 2016 and from 18th May 2016 to 18th Apr 2017 on 624 days (66.10 % availability). The logger located in the salt marsh enclosed plot (S3Pio) recorded water conductivity from 07th May 2015 to 08th Oct 2015, 03rd Dec 2015 to 16th Feb 2016 and from 18th

May 2016 to 18th Apr 2017 in total 567 days (60.06 % availability). Gaps in the dataset resulted from a malfunction and maintenance.



## 4 Results and discussion

### 4.1 Weather Station (WS)

Local wind roses for three winter storm seasons and one summer season are shown in figure 5. Furthermore, minima, maxima and mean values as well as median and standard deviation of the Weather Station (WS) for the time frame of 19th Nov 2014 to 18th Apr 2017 on 571 days are listed in table 10. Within the application time, wind speeds of less than 25 m /s were present. Mean wind speed was 5.7 m/s ± 3.21 but it strongly differ between the single seasons.

### 4.2 Recording Current Meter (RCM)

Current conditions for one storm season and one summer season are shown in figure 6. A main current direction from southwest to northeast were clearly identified showing the good orientation of the experimental islands. Furthermore, minima, maxima and mean values as well as median and standard deviation of the RCM data from 18th Sep 2014 until 06th Oct 2015 on 335 days are listed in table 11. A maximum current speed of 107.05 cm/s could be observed in Dec 2013 during storm Xaver. Mean current speed represent 12.9 cm/s ± 8.62, but with variances between the single seasons.

### 4.3 Tide and Wave Recorder (TWR)

Water depth, calculated from pressure data of the Tide and Wave Recorder (TWR) can be seen in figure 7. Variability during seasons are obviously for some parameters and apparently highest water depth were reached during storm season. However, water level reached the upper salt marsh zone (2.0 m NHN) several times (e.g. Apr 2015, Jul 2015, Aug 2016). During the storms Elon and Felix in Jan 2015 highest water level was observed with 3.62 m as well as the highest wave with 2.14 m (Fig. 8). Anyhow, the highest wave energy (400.90 J/m²) and significant wave height H1/3 (0.73 m) were detected during several shortly sequenced storm events the weeks before in Dec 2014 (Fig. 8). Mean water depths for the whole application are 1.23 m ±0.35 m. Statistic values of the TWR from 01st Oct 2014 to 18th Apr 2017 on 812 days are listed in table 12.

### 4.4 Data Logger (DL)

### 4.4.1 Water Level Loggers (DL-W)

Depth of groundwater level achieved from pressure logger inside the experimental islands and salt marsh enclosed plots can be seen in figure 9. Differences in the water level have been observed especially during low water. This could be a result of various factors ranging from diverse water consumption of plants, less flooding on higher elevations or leaks in the plastic bags inside an experimental island. Statistics of ground water level data (DL-W 1-6) for each logger application time from 20th Jun 2015 to 18th Apr 2017 on 578 days (572 days within the salt marsh enclosed plot) are listed in table 13.



### 4.4.2 Temperature Loggers (DL-T)

Surface layer temperatures for one experimental island and its three elevational zones as well as one elevation zone within the salt marsh enclosed plots are shown in figure 10. As a basis of temperature differences the DL-T2 Logger (I3 Low) was taken to compare both differences within one islands elevational zones (DL-T1 I3 Pio, DL-T3 I3 Upp) and between the same elevational level compared to one of the salt marsh control plots (DL-T4, S3 Low). All temperatures are very similar and showing only less differences, especially I3 Upp and I3 Low with ΔT 0.01 ± 0.55 °C. Since the temperatures within the experimental islands are very similar considering the mean monthly temperatures of the four plots a clear offset of the salt marsh enclosed plot temperature could be identified in winter and summer month (Fig. 11). Lowest Temperature was observed at I3 Pio with -2.6 °C whereas the highest value with 30.43 °C was measured at the upper zone of the experimental island. Maximum temperatures of S3 Low and I3 Upp differ 9.31 °C. Mean values are very similar and reaching from 9.64°C ± 5.02 (S3 Low) to 9.78 ± 5.74 (I3 Pio). More statistics of temperature data (DL-L 1-6) for each logger application time from 18th Sep 2014 until 18th Apr 2017 on 931 days are listed in table 14.

### 4.4.3 Light Loggers (DL-L)

To calculate light sum per day measured light intensity (Quantum [μmol / m²s]) was extrapolated from seconds to the measuring interval of 10 minutes and afterwards values were added up for one day. As storm events were identified as a source for scouring processes especially in the bare islands (Balke et al. 2017) polycarbonate covers were installed for the experimental islands during storm season beginning in Oct 2015. In consequence, lower light availabilities were detected under the covers (Fig. 12) at island 3 in I3 Pio, I3 Low and I3 Upp other than at the seafloor, I3 Pole and S 3 Low (all not covered). Minima, maxima and mean values as well as median and standard deviation of the light intensity data (DL-L 1-6) for each logger application over the whole period from 18th Sep 2014 to 18th Apr 2017 for 931 days are listed in table 15.

### 4.4.4 Conductivity Loggers (DL-C)

Salinity values achieved from both conductivity logger within the experimental island and the salt marsh enclosed plot can be seen in figure 13. Due to fluctuations in the ground water level conductivity loggers periodically fell dry especially in the beginning. Thus, data until Oct 2015 are scattered. Thereupon the depth of conductivity loggers were adjusted to deeper in the dip wells assuring a constant covering of water. Mean salinity for the experimental island is 27.95 ppt ± 1.22 and for the salt marsh enclosed plot 25.45 ppt ± 1.74. Both logger were installed at the pioneer zone. Statistics of salinities (DL-C 1-2) for each logger application time from 06th May 2015 to 16th Feb 2016 and from 18th May 2016 to 18th Apr 2017 on 624 days are shown in table 16.



## 5 Conclusions and future directions

The BEFmate project included a variety of experiments dedicated to investigate biodiversity – ecosystem functioning across marine and terrestrial ecosystems utilizing among others 12 artificial islands in the back-barrier tidal-flat of Spiekeroog Island. Abiotic conditions were recorded from a suite of 23 different sensors installed at different locations in the vicinity of the

5 experimental islands, within the islands themselves and within the nearby salt marsh. Data described here covers the period from Sep 2014 to Apr 2017 and has been published in seven datasets in the World Data Center PANGAEA. Data coverage within the period reached from 35% for the recording current meter (RCM) that failed in October 2015, to 99% for 6 data loggers. With 17 sensors covering 80% or more of the period of interest, a very good coverage was achieved. Additional data from pre-experiment investigations with the RCM between November 2013 and June 2017 was added to the dataset. Seasonal

and tidal dynamics, as well as storms were covered and these data are available for interpretation in further contexts.

For future operations data availability can be further increased if a rotational system for maintenance is applied, providing spare sensors are available. Furthermore, online data transfer of central information would not only increase the data availability, but meteorological and hydrographic conditions can indicate the need to attend the experiment. For example, indicating a need to take actions to prevent damage during extreme weather events or allow us to plan event based sampling.

Non-invasive remote sensing sensor techniques can provide complementary data, to avoid fouling issues, as demonstrated successfully at the nearby Time Series Station (TSS) Spiekeroog (Garaba et al., 2014; Schulz et al. 2016). Additionally camera systems should be applied providing a visual impression of the overall scene and detailed information, e.g. the process of flooding for different level. Recently the RCM failure within the BEFmate project has inspired the development of a machine-learning environment that creates a virtual sensor enabling to compensate for single sensor dropouts (Oehmcke et al. 2017a,

2017b). Finally, data quality assurance and quality measures should be further developed to reduce the workload of manual data curation while improving data availability in near-real time.

### Sample availability

The dataset described within this document is based on sensor information only, that had been uploaded to the PANGAEA database (see section 2.3).

**Appendices**

None

**Supplement link**

None



**Author contribution**

Oliver Zielinski was responsible for the scientific approach and performed the interpretation of results. He drafted and prepared the manuscript.

Daniela Meier was responsible for technical maintenance, laboratory analysis, performed fieldwork, data sampling and
analysis, created figures and contributed to the writing process of the manuscript.

Helmut Hillebrand was principle investigator of the BEFmate project and contributed to the writing process.

Michael Kleyer was principle investigator of one of the BEFmate subprojects and responsible for the conception of the experimental islands.

Thorsten Balke and Kertu Lõhmus were responsible for maintenance of the experimental islands as well as fieldwork
coordination.

All authors contributed to proof reading of the manuscript.

**Competing interests**

The authors declare that they have no conflict of interest.

**Acknowledgements**

The authors are very grateful to Kai Schwalfenberg and Claudia Thölen for their tremendous support in the field and laboratory. Sincere thanks to Daniela Voß, Ursel Gerken, Kathrin Dietrich, Nick Rüssmeier, Rohan Henkel, Jule Beßler, Franziska Wöhrmann and Hauke Haake for technical, laboratory or fieldwork support. Special thanks to Helmo Nicolai and Gerrit Behrens for their technical and logistical support. The support and cooperation with Nationalparkverwaltung Niedersächsisches Wattenmeer and the Umweltzentrum Wittbülten is acknowledged as well as the support of Rainer Sieger at
PANGAEA. Thanks to the BEFmate colleagues and all other helping hands in the field during sampling campaigns, especially Regine Redelstein. The BEFmate project (Biodiversity - Ecosystem Functioning across marine and terrestrial ecosystems) was funded by the Ministry for Science and Culture of Lower Saxony, Germany under project number ZN2930.

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



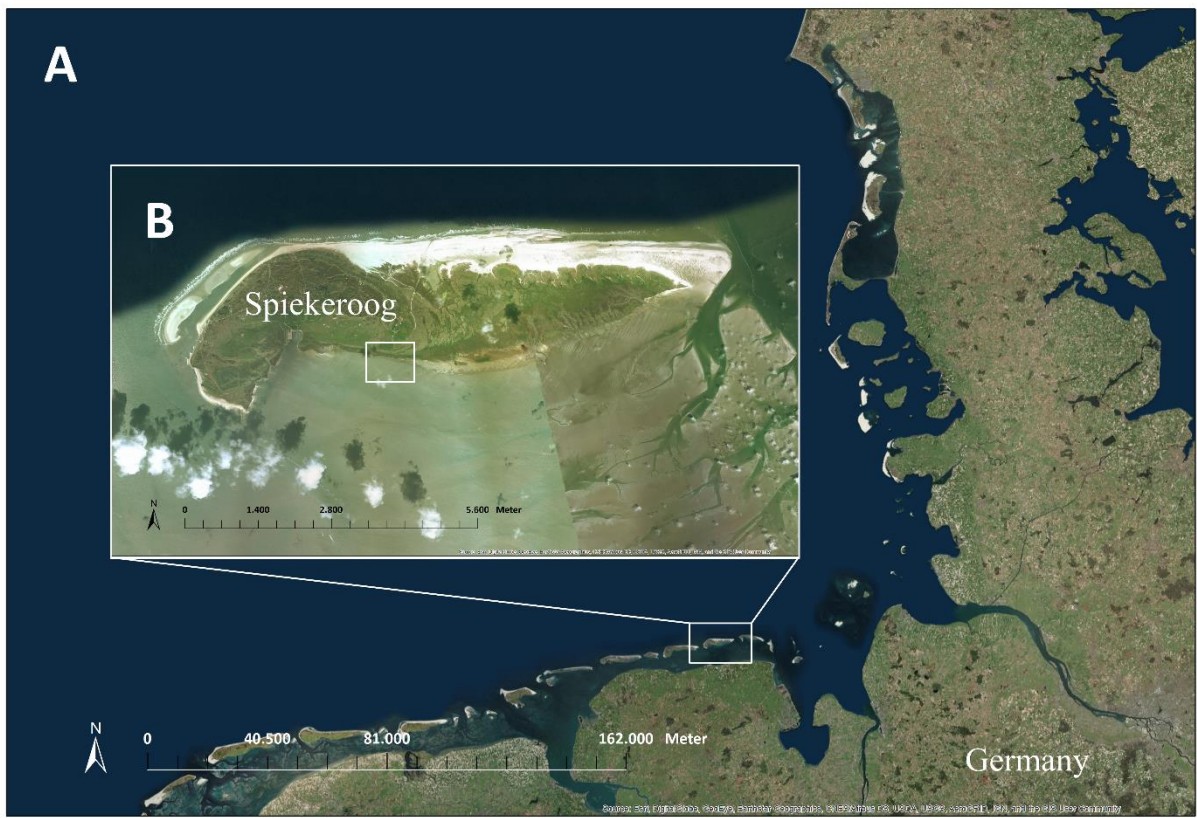

**Figure 1: (A) Island of Spiekeroog located in the German Bight (Southern North Sea). (B) The study site is located south of Spiekeroog (see insert) in the back-barrier tidal flat for experimental islands and in the adjacent salt marsh for salt marsh enclosed plots (white box).**



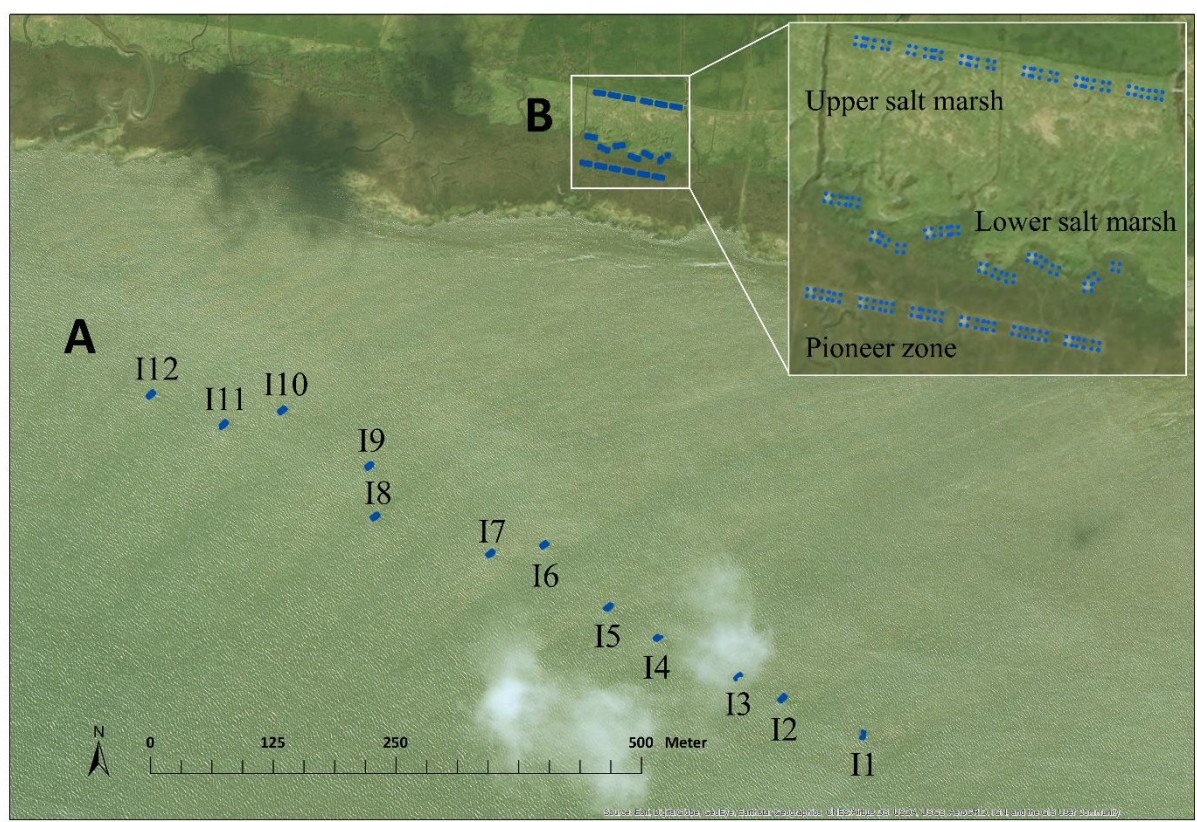

**Figure 2: Experiment setup with (A) experimental islands in the back-barrier tidal flat and (B) salt marsh control plots at different elevational zones in the salt marsh. Both, island and salt marsh experimental plots are numbered from 1 to 12 from east to west.**

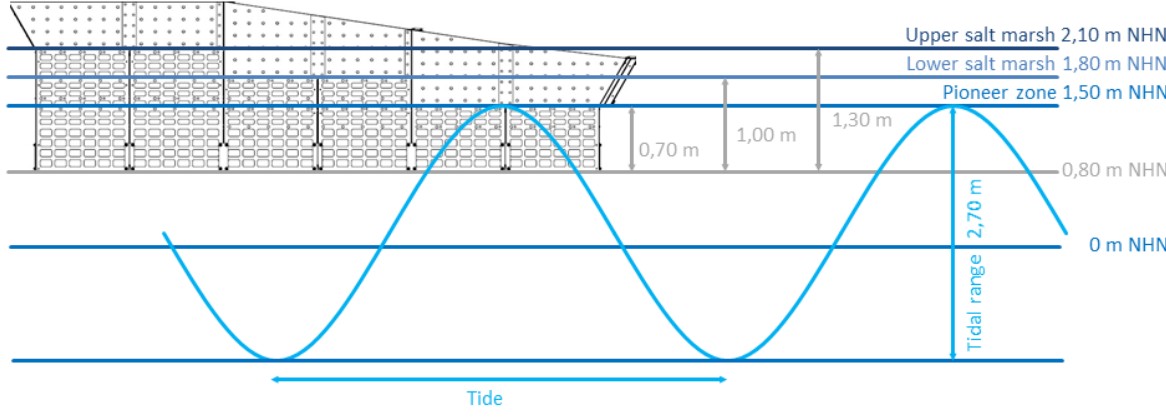

**Figure 3: Scheme of an experimental island showing three elevational levels representing natural salt marsh zones. The islands are built on average at 0.8 m NHN.**





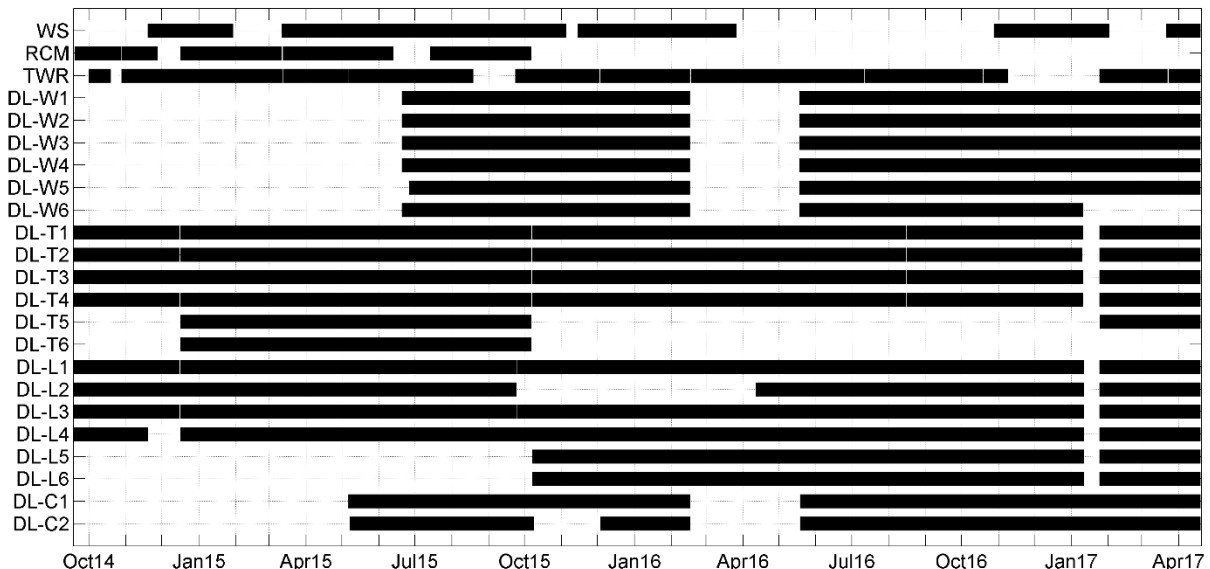

Figure 4: Available data shown in black bars for each sensor or logger type over the sampling period of 18th Sep 2014 to 18th Apr 2017.





**Figure 5: Local wind roses (wind speed and wind direction) for storm seasons (01st Oct - 31st Mar) in 2014/2015 (A), 2015/2016 (C) and 2016/2017 (D) as well as for one summer season (01st Apr - 30th Sep) in 2015 (B).**




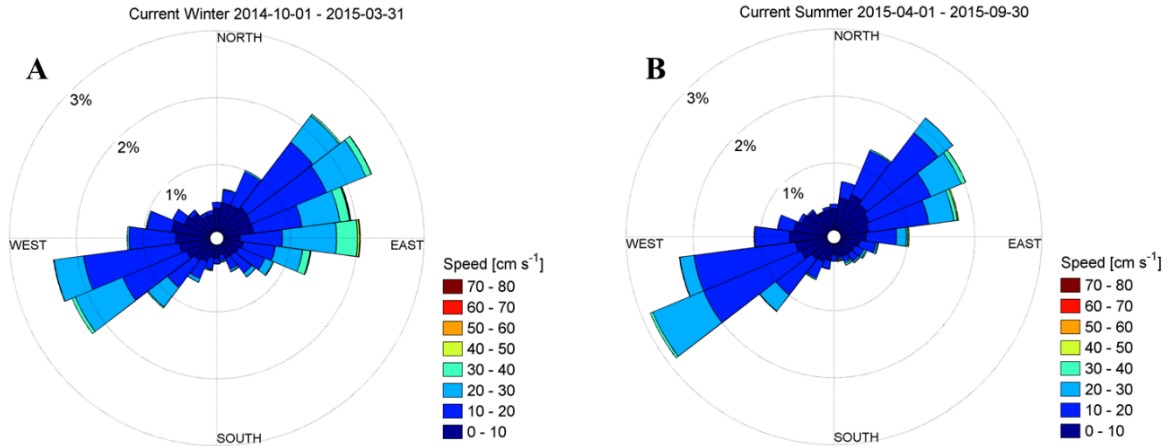

**Figure 6: Current conditions in storm season (left) from Oct 2014 to Mar 2015 and summer season (right) from Apr 2015 to Sep 2015 with a main current direction from southwest to northeast.**

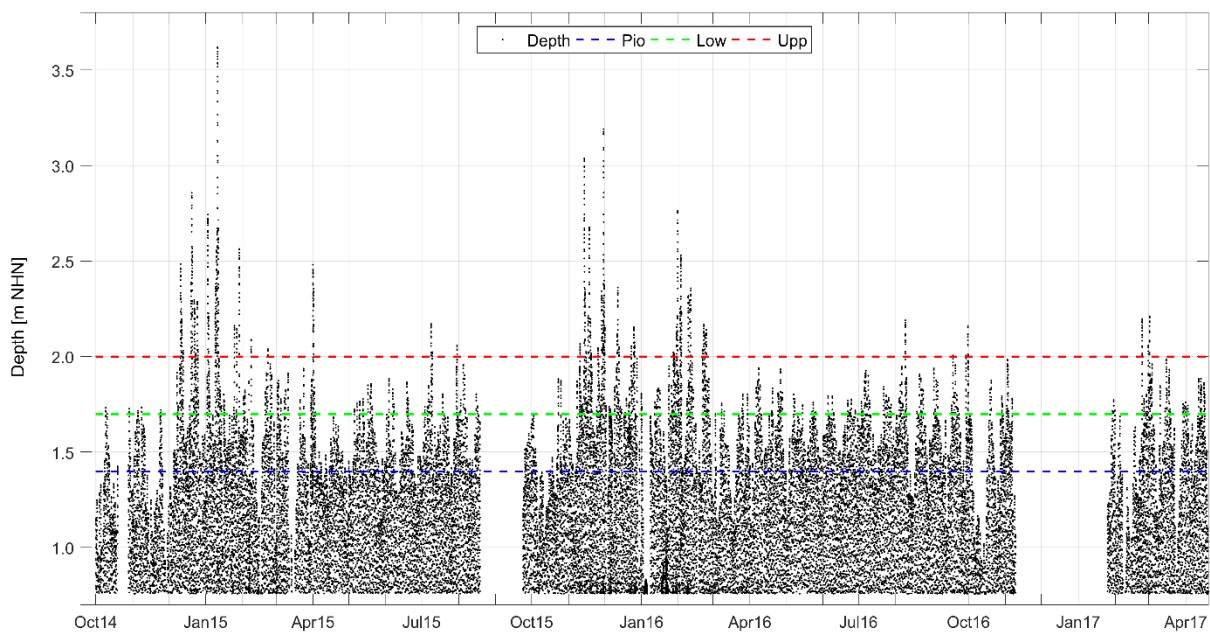

**Figure 7: Calculated water depth (black dots) at 0.71 m NHN next to island 6 and 7. The blue line represents the elevation of the pioneer zone, the green line is showing the height of the lower salt marsh and the red line describes the upper salt marsh zone. Thus, water level data can give information of flooding periods at the three elevation levels. Low tide data (all data below 0.05 m) was removed before plotting the data. To provide m NHN as a basis all data points 0.71 m were added to the data points.**



**Figure 8: Calculated water depth (black line) during different weather conditions. A shows regular weather and wave conditions from 16th to 22nd Jan 2015. B represent a series of storms with higher water levels from 19th until 25th Dec 2015. At this point**



highest wave energy values were observed as well as highest significant wave heights. C shows the two big storms Elon and Felix which occurred directly one after the other. Here the highest water level during application time were measured as well as the maximum wave height.

Figure 9: Ground water level within the experimental island and the salt marsh enclosed plot.



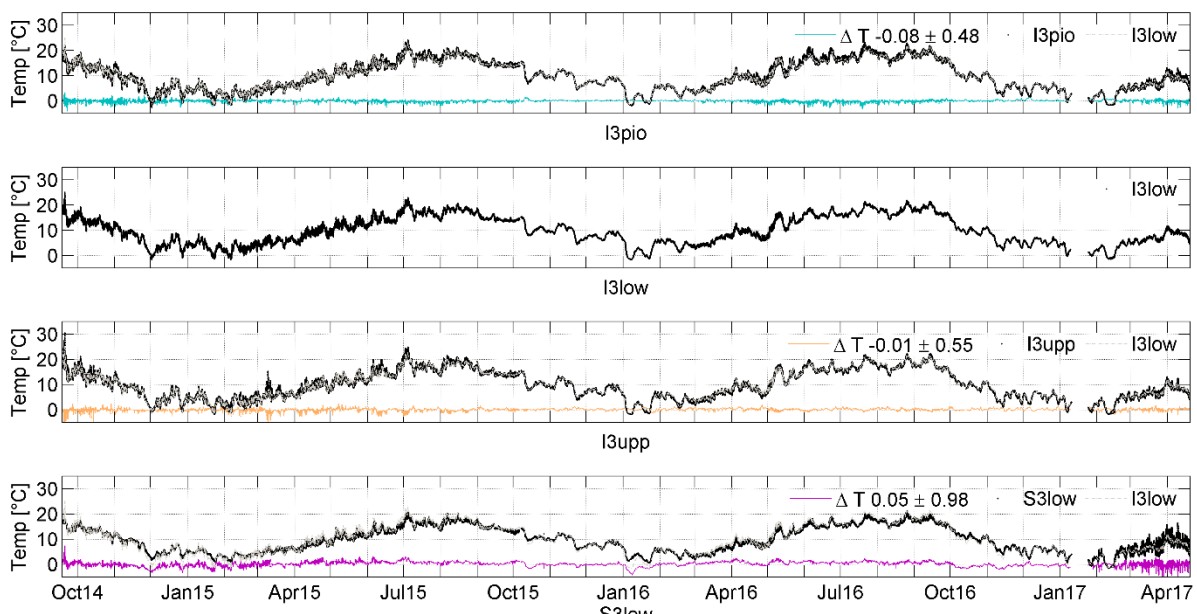

**Figure 10: Surface layer temperature over the whole time frame with the DL-T2 (I3 Low) as a basis of computed temperature differences.**

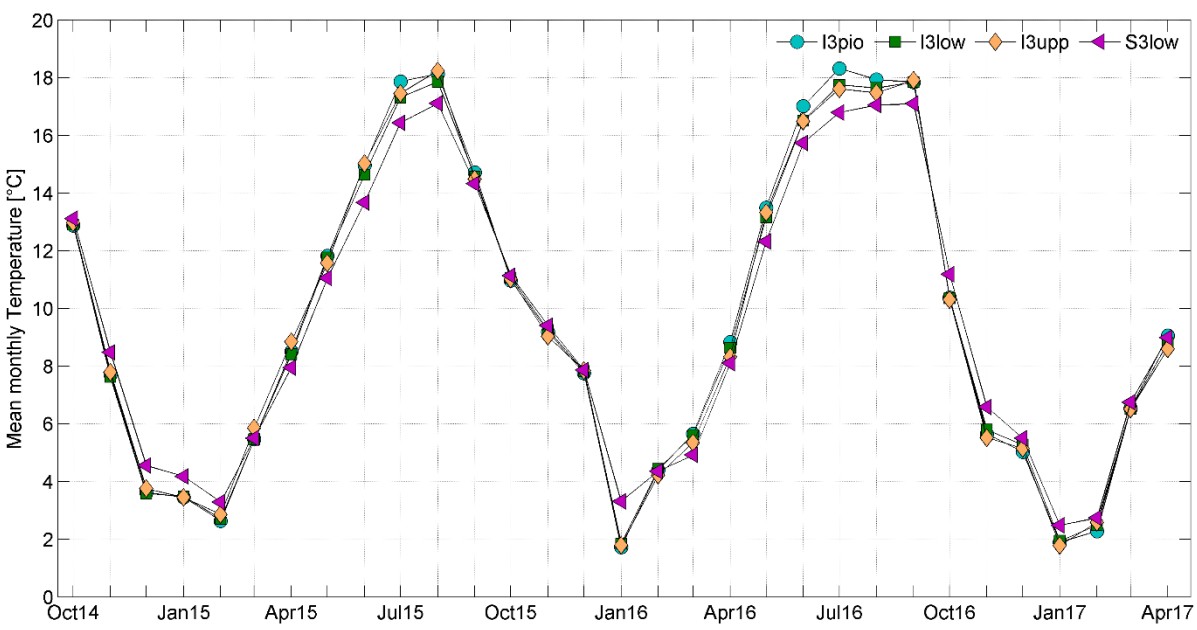

**Figure 11: Mean monthly temperature over the whole application time. Since the temperatures within the experimental islands are very similar, a clear offset of the salt marsh enclosed plot temperature could be identified in winter and summer month.**





**Figure 12: Calculated sum of light intensity per day.**





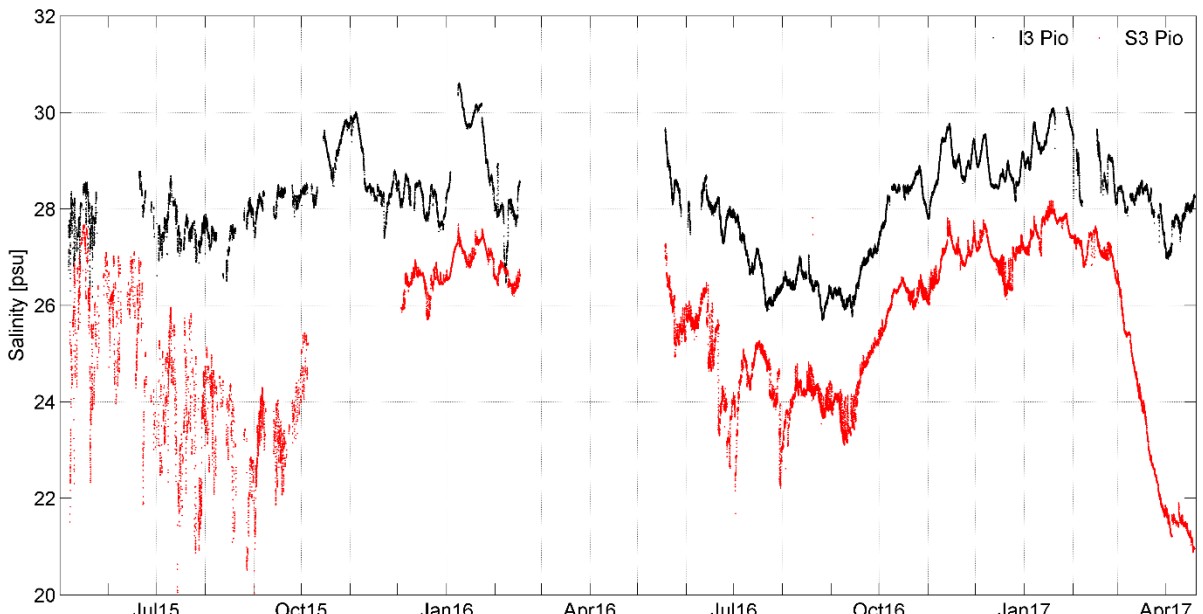

**Figure 13: Salinity values within dip wells inside the experimental island and salt marsh enclosed plot. Due to fluctuations in the ground water level conductivity loggers periodically fell dry especially in the beginning. Thus, data until October 2015 are scattered. Thereupon the depth of conductivity loggers were adjusted to deeper in the dip wells assuring a constant covering of water.**

**Table 1: Overview of all installed loggers at the experimental island (I) and the salt marsh enclosed plots (S) as well as the nearby installed sensors.**

| Logger | Name | Location | Elevation | Latitude | Longitude | Height [m NHN] | Parameter |
|---|---|---|---|---|---|---|---|
| WS | WS | Shore | | 53°45'57.10"N | 7°43'34.11"E | 10 m | Meteorology |
| RCM | RCM | Seafloor | | 53°45'29.34"N | 7°43'16.50"E | 0.8 m | Current |
| TWR | RBR | Seafloor | | 53°45'29.34"N | 7°43'16.50"E | 0.8 m | Tide/Wave |
| DL-W0 | Hobo-P | I 3 | Pole | 53°45'26.33"N | 7°43'29.16"E | 3.5 m | AtmPressure |
| DL-W1 | Hobo-P | I 3 | Upp | 53°45'26.33"N | 7°43'29.16"E | 1.0 m | Depth |
| DL-W2 | Hobo-P | I 3 | Low | 53°45'26.38"N | 7°43'29.24"E | 0.9 m | Depth |
| DL-W3 | Hobo-P | I 4 | Pio | 53°45'27.68"N | 7°43'24.85"E | 1.0 m | Depth |
| DL-W4 | Hobo-P | I 1 | Pio | 53°45'24.61"N | 7°43'36.26"E | 1.1 m | Depth |
| DL-W5 | Hobo-P | S 3 | Low | 53°45'43.62"N | 7°43'23.91"E | 0.9 m | Depth |
| DL-W6 | Defi-D | I 4 | Low | 53°45'27.64"N | 7°43'24.75"E | 1.0 m | Depth |
| DL-T1 | Defi-T | I 3 | Pio | 53°45'26.42"N | 7°43'29.34"E | 1.4 m | Temperature |
| DL-T2 | Defi-T | I 3 | Low | 53°45'26.38"N | 7°43'29.24"E | 1.7 m | Temperature |



| DL-T3 | Defi-T | I 3 | Upp | 53°45'26.33"N | 7°43'29.16"E | 2.0 m | Temperature |
|---|---|---|---|---|---|---|---|
| DL-T4 | Defi-T | S 3 | Low | 53°45'43.62"N | 7°43'23.91"E | 1.7 m | Temperature |
| DL-T5 | Defi-T | S 2 | Upp | 53°45'45.17"N | 7°43'24.80"E | 2.0 m | Temperature |
| DL-T6 | Defi-T | S 2 | Low | 53°45'43.54"N | 7°43'24.27"E | 1.7 m | Temperature |
| DL-L1 | Defi-L | I 3 | Pio | 53°45'26.42"N | 7°43'29.34"E | 1.4 m | Light |
| DL-L2 | Defi-L | I 3 | Low | 53°45'26.38"N | 7°43'29.24"E | 1.7 m | Light |
| DL-L3 | Defi-L | I 3 | Upp | 53°45'26.33"N | 7°43'29.16"E | 2.0 m | Light |
| DL-L4 | Defi-L | S 3 | Low | 53°45'43.62"N | 7°43'23.91"E | 3.5 m | Light |
| DL-L5 | Defi-L | I 3 | Pole | 53°45'26.33"N | 7°43'29.16"E | 2.5 m | Light |
| DL-L6 | Defi-L | Seafloor | | 53°45'29.34"N | 7°43'16.50"E | 0.8 m | Light |
| DL-C1 | Hobo-C | I 3 | Pio | 53°45'26.42"N | 7°43'29.34"E | 0.9 m | Salinity |
| DL-C2 | Hobo-C | S 3 | Pio | 53°45'42.88"N | 7°43'23.55"E | 0.9 m | Salinity |

**Table 2: Structure of the Weather Station (WS) datasets at PANGAEA.**

| # | Name | Short Name | Unit |
|---|---|---|---|
| 1 | DATE/TIME | Date/Time | |
| 2 | ALTITUDE | Altitude | m |
| 3 | Wind speed | ff | m/s |
| 4 | Wind direction | dd | deg |
| 5 | Temperature, air | TTT | degC |
| 6 | Humidity, relative | RH | % |
| 7 | Brightness, North | Brightness | lux |
| 8 | Brightness, East | Brightness | lux |
| 9 | Brightness, South | Brightness | lux |
| 10 | Brightness, West | Brightness | lux |
| 11 | Brightness, Max | Brightness | lux |
| 12 | Brightness direction | Brightness dir | deg |
| 13 | Indicator for inclusion or omission of precipitation data | Indicator precipitation | code |
| 14 | Precipitation, daily total | Precip day total | mm |
| 15 | Precipitation | Precip | mm/h |
| 16 | Precipitation description | Precip descry | |
| 17 | Solar elevation | SE | deg |



| 18 | Solar azimuth angle | SAA | deg |
|----|---------------------|-----|-----|

**Table 3: Structure of the Recording Current Meter (RCM) datasets at PANGAEA.**

| # | Name | Short Name | Unit |
|---|------|------------|------|
| 1 | DATE/TIME | Date/Time | |
| 2 | Depth, water | Depth water | m |
| 3 | Current speed | V | cm/s |
| 4 | Current direction | DIR | deg |
| 5 | Temperature, water | Temp | degC |
| 6 | Conductivity | Cond | mS/cm |
| 7 | Pressure, raw | Press raw | dbar |

**Table 4: Structure of the Tide and Wave Recorder (TWR) datasets PANGAEA.**

| # | Name | Short Name | Unit |
|---|------|------------|------|
| 1 | DATE/TIME | Date/Time | |
| 2 | Depth, water | Depth water | m |
| 3 | Tidal slope | Tidal slope | |
| 4 | Wave height, significant | H(1/3) | m |
| 5 | Wave period, significant | T(1/3) | s |
| 6 | Wave height, tenth | H(1/10) | m |
| 7 | Wave period, tenth | T(1/10) | s |
| 8 | Wave height, maximum | H(max) | m |
| 9 | Wave period, maximum | T(max) | s |
| 10 | Wave height, average | H(avg) | m |
| 11 | Wave period, average | T(avg) | s |
| 12 | Wave energy | Wave energy | J/m² |
| 13 | Temperature, water | Temp | degC |
| 14 | Pressure, raw | Press raw | dbar |

**Table 5: Structure of the Water Level Loggers (DL-W) datasets at PANGAEA.**

| # | Name | Short Name | Unit |
|---|------|------------|------|
| 1 | DATE/TIME | Date/Time | |





| 2 | Water level | Water level | m |
|---|---|---|---|

**Table 6: Structure of the Temperature Loggers (DL-T) datasets at PANGAEA.**

| # | Name | Short Name | Unit |
|---|---|---|---|
| 1 | DATE/TIME | Date/Time | |
| 2 | Depth, sediment/rock | Depth | m |
| 3 | Temperature, in rock/sediment | t | degC |

**Table 7: Structure of Light Loggers (DL-L) datasets at PANGAEA.**

| # | Name | Short Name | Unit |
|---|---|---|---|
| 1 | DATE/TIME | Date/Time | |
| 2 | Light intensity | Io | µmol/m²/s |

**Table 8: Structure of Conductivity Logger (DL-C) datasets at PANGAEA.**

| # | Name | Short Name | Unit |
|---|---|---|---|
| 1 | DATE/TIME | Date/Time | |
| 2 | Depth, sediment/rock | Depth | m |
| 3 | Salinity | Sal | |

**Table 9: An overview of sensor and logger data availability over the 944-day sampling period. D 1 – sum of days with data available; D 0 – sum of days with data absent; % 1 – proportion of days with data available; % 0 – proportion of days with data absent; D x - total available days specified for each sensor application, i.e. from first deployment to last measurement; % x1 and % x0 are the corresponding percent availabilities. Sum 1 – number of measurements available; Sum 0 – number of measurements missing; % S 1 - proportion of measurements available; % S 0 - proportion of measurements missing; Sum x – total available measurements specified of each sensor application, i.e. from first data point to last measurement; %S x1 and %S x0 are the corresponding percent availabilities.**

| | WS | RCM | TWR | DL-W1 | DL-W2 | DL-W3 | DL-W4 | DL-W5 | DL-W6 | DL-C1 | DL-C2 | DL-T1 |
|---|---|---|---|---|---|---|---|---|---|---|---|---|
| | | Sea-floor | Sea-floor | I4 Pio | I3 Upp | I3 Low | I1 Pio | S3 Low | I4 Low | I3 Pio | S3 Pio | I3 Pio |
| D 1 | 571 | 335 | 812 | 578 | 578 | 578 | 578 | 572 | 480 | 624 | 567 | 931 |
| D 0 | 373 | 50 | 132 | 366 | 366 | 366 | 366 | 372 | 464 | 320 | 377 | 13 |
| % 1 | 60,49 | 35,49 | 86,02 | 61,23 | 61,23 | 61,23 | 61,23 | 60,59 | 50,85 | 66,10 | 60,06 | 98,62 |
| % 0 | 39,51 | 5,30 | 13,98 | 38,77 | 38,77 | 38,77 | 38,77 | 39,41 | 49,15 | 33,90 | 39,94 | 1,38 |

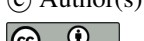



| D x | 882 | 384 | 931 | 669 | 669 | 669 | 669 | 663 | 571 | 714 | 713 | 944 |
|---|---|---|---|---|---|---|---|---|---|---|---|---|
| % x1 | 64,74 | 87,24 | 87,22 | 86,40 | 86,40 | 86,40 | 86,40 | 86,27 | 84,06 | 87,39 | 79,52 | 98,62 |
| % x0 | 35,26 | 12,76 | 12,78 | 13,60 | 13,60 | 13,60 | 13,60 | 13,73 | 15,94 | 12,61 | 20,48 | 1,38 |
| | | | | | | | | | | | | |
| Sum 1 | 816050 | 47490 | 115535 | 82997 | 82999 | 83000 | 83002 | 82112 | 68850 | 89481 | 81297 | 133311 |
| Sum 0 | 542410 | 88356 | 20311 | 52849 | 52847 | 52846 | 52844 | 53734 | 66996 | 46365 | 54549 | 2535 |
| %S 1 | 60,07 | 34,96 | 85,05 | 61,10 | 61,10 | 61,10 | 61,10 | 60,44 | 50,68 | 65,87 | 59,84 | 98,13 |
| %S 0 | 39,93 | 65,04 | 14,95 | 38,90 | 38,90 | 38,90 | 38,90 | 39,56 | 49,32 | 34,13 | 40,16 | 1,87 |
| Sum x | 1269390 | 55093 | 134064 | 96288 | 96288 | 96288 | 96288 | 95407 | 82118 | 102745 | 102613 | 135846 |
| %S x1 | 64,29 | 86,20 | 86,18 | 86,20 | 86,20 | 86,20 | 86,20 | 86,06 | 83,84 | 87,09 | 79,23 | 98,13 |
| %S x0 | 35,71 | 13,80 | 13,82 | 13,80 | 13,80 | 13,80 | 13,80 | 13,94 | 16,16 | 12,91 | 20,77 | 1,87 |

**Table 10: Minima, maxima and mean values as well as median and standard deviation of the weather station (WS) for the time frame of 19th Nov 2014 to 18th Apr 2017 on 571 days.**

| WS | Min | Max | Mean | Median | SD |
|---|---|---|---|---|---|
| ff [m/s] | 0,00 | 24,68 | 5,70 | 5,04 | 3,21 |
| dd [°] | 0,00 | 379,99 | 197,52 | 210,02 | 87,20 |
| TTT [°C] | -8,10 | 32,20 | 8,23 | 7,50 | 5,57 |
| RH [%] | 2,00 | 100,00 | 87,75 | 90,00 | 10,37 |
| PPPP [hPa] | 977,58 | 1042,07 | 1018,12 | 1018,68 | 10,37 |
| Brightness North [lux] | 0,00 | 89502,54 | 4785,86 | 6,23 | 9441,66 |
| Brightness East [lux] | 0,00 | 149582,40 | 8520,22 | 6,59 | 19157,07 |
| Brightness South [lux] | 0,00 | 150000,00 | 9685,35 | 6,41 | 21869,61 |
| Brightness West[lux] | 0,00 | 143925,20 | 6701,04 | 6,90 | 15295,02 |
| Brightness Max. [lux] | 0,00 | 150000,00 | 12376,23 | 8,40 | 25729,80 |
| Brightness dir [°] | 0,00 | 360,00 | 71,71 | 0,00 | 101,66 |
| Precip day total [mm] | 0,00 | 6,00 | 0,00 | 0,00 | 0,01 |
| Precip [mm/h] | 0,00 | 68,00 | 0,14 | 0,00 | 1,00 |
| SE [°] | -60,40 | 60,40 | -4,75 | -4,06 | 29,43 |
| SAA [°] | 0,00 | 359,83 | 180,00 | 180,07 | 100,02 |

5    **Table 11: Minima, maxima and mean values as well as median and standard deviation of the Recording Current Meter (RCM) for its application time from 18th Sep 2014 until 06th Oct 2015 on 335 days.**




| RCM | Min | Max | Mean | Median | SD |
|---|---|---|---|---|---|
| Speed [cm/s] | 0,29 | 107,05 | 14,09 | 12,90 | 8,62 |
| Direction [°] | 0,00 | 359,70 | 157,41 | 134,66 | 98,54 |
| Temperature [°C] | -0,68 | 27,30 | 10,69 | 10,25 | 5,43 |
| Conductivity [mS/cm] | 24,39 | 50,60 | 33,82 | 32,08 | 4,97 |
| Pressure [kPa] | 93,74 | 136,71 | 103,78 | 101,60 | 4,03 |

**Table 12: Minima, maxima and mean values as well as median and standard deviation of the Tide and Wave Recorder (TWR) from 01st Oct 2014 to 18th Apr 2017 on 812 days.**

| TWR | Min | Max | Mean | Median | SD |
|---|---|---|---|---|---|
| Depth [m] | 0,69 | 3,62 | 1,23 | 1,19 | 0,35 |
| Tdslope | -1,48 | 1,08 | 0,00 | 0,00 | 0,21 |
| H(1/3) [m] | 0,00 | 0,73 | 0,03 | 0,00 | 0,06 |
| T(1/3) [s] | 0,00 | 271,00 | 3,58 | 1,66 | 9,60 |
| H(1/10) [m] | 0,00 | 0,96 | 0,04 | 0,00 | 0,08 |
| T(1/10) [s] | 0,00 | 272,67 | 5,47 | 1,68 | 14,12 |
| H(max) [m] | 0,00 | 2,14 | 0,05 | 0,00 | 0,10 |
| T(max) [s] | 0,00 | 288,33 | 7,97 | 1,67 | 20,29 |
| H(avg) [m] | 0,00 | 0,51 | 0,02 | 0,00 | 0,04 |
| T(avg) [s] | 0,00 | 288,33 | 2,01 | 1,46 | 4,84 |
| Energy [J/m²] | 0,00 | 400,90 | 3,67 | 0,00 | 14,02 |
| Temp [°C] | -6,10 | 31,87 | 10,00 | 9,29 | 6,20 |
| Pressure raw [dbar] | 9,71 | 12,98 | 10,36 | 10,25 | 0,33 |

5   **Table 13: Minima, maxima and mean values as well as median and standard deviation of the ground water level data (DL-W 1-6) for each logger application time from 20th Jun 2015 to 18th Apr 2017 on 578 days (572 days within the salt marsh enclosed plot).**

| DL-W | Min | Max | Mean | Median | SD |
|---|---|---|---|---|---|
| DL-W1 Water level I3upp [m] | 1,03 | 3,44 | 1,33 | 1,30 | 0,15 |
| DL-W2 Water level I3low [m] | 0,88 | 3,34 | 1,28 | 1,27 | 0,27 |
| DL-W3 Water level I4pio [m] | 0,93 | 3,56 | 1,45 | 1,48 | 0,20 |
| DL-W4 Water level I1pio [m] | 0,78 | 3,39 | 1,25 | 1,23 | 0,17 |
| DL-W5 Water level S3low [m] | 0,87 | 3,38 | 1,39 | 1,35 | 0,19 |
| DL-W6 Water level I4low [m] | 0,64 | 3,43 | 1,35 | 1,33 | 0,20 |



**Table 14: Minima, maxima and mean values as well as median and standard deviation of the temperature data (DL-T 1-6) from 18th Sep 2014 until 18th Apr 2017 on 931 daysTable.**

| DL-T | Min | Max | Mean | Median | SD |
|---|---|---|---|---|---|
| DL-T1 Temperature I3 Pio [°C] | -2,60 | 24,14 | 9,78 | 9,12 | 5,74 |
| DL-T2 Temperature I3 Low [°C] | -1,75 | 25,09 | 9,70 | 9,05 | 5,56 |
| DL-T3 Temperature I3 Upp [°C] | -1,87 | 30,43 | 9,70 | 9,07 | 5,66 |
| DL-T4 Temperature S3 Low [°C] | -1,52 | 21,12 | 9,64 | 9,16 | 5,02 |

5 **Table 15: Minima, maxima and mean values as well as median and standard deviation of the light intensity data (DL-L 1-6) for each logger application time and separated in years (2014 – 2017).**

| DL-L | Min | Max | Mean | Median | SD |
|---|---|---|---|---|---|
| DL-L1 Light I3pio [µmol/(m²s)] | 0,00 | 2774,60 | 105,16 | 0,50 | 291,63 |
| DL-L2 Light I3low [µmol/(m²s)] | -0,40 | 2603,30 | 120,22 | 1,10 | 301,43 |
| DL-L3 Light I3upp [µmol/(m²s)] | 0,00 | 2609,70 | 141,10 | 0,80 | 343,06 |
| DL-L4 Light S3low [µmol/(m²s)] | 0,20 | 2657,80 | 244,58 | 4,20 | 435,98 |
| DL-L5 Light I3upp pole [µmol/(m²s)] | -0,30 | 2706,30 | 222,82 | 1,10 | 412,30 |
| DL-L6 Light sea floor [µmol/(m²s)] | 0,00 | 2058,10 | 85,15 | 0,50 | 196,23 |

**Table 16: Minima, maxima and mean values as well as median and standard deviation of the salinity data (DL-C 1-2) for each logger application time and separated in years (2015 – 2017).**

| DL-C | Min | Max | Mean | Median | SD |
|---|---|---|---|---|---|
| DL-C1 Sal I3pio [ppt] | 20,00 | 30,61 | 27,95 | 28,17 | 1,22 |
| DL-C2 Sal S3pio [ppt] | 20,04 | 28,17 | 25,45 | 25,91 | 1,74 |

