# Peer review of "Environmental conditions of a salt-marsh biodiversity experiment on the island of Spiekeroog (Germany)"

_Earth System Science Data, 2018_

## Referee Comment (RC1) · J. Piera (Referee) · 9 Aug 2018

In this contribution the authors report the abiotic parameters observed from 23 sensors installed around the facilities installed in the framework of the project BEFmate.

My major concern is about the data quality procedures. According to the method described by the authors, quality control was performed by a) erasing negative readings and data covering maintenance activities, b) visual inspection of the overall dataset and c) removal of outliers, defined as data exhibiting changes of more than two standard deviations within one time step.

[Figure]

Although the authors mention that data quality assurance and quality measures will be further developed, this primary quality control has some potential problems: "Visual inspection of the overall dataset" is always a fuzzy concept that provably change the criteria among different sensors. Furthermore, it is not reproducible.

At present there are several publications and guidelines to apply widely accepted criteria for data quality control of oceanographic data. See for example the guideline from SeaDataNet: https://www.seadatanet.org/Standards/Data-Quality-Control

I suggest also to keep the original data and provide quality flags. It could be recommendable also to include a script to perform the quality actions to apply to the data (delete, interpolate, ...) based on the quality flag information. Using this approach, other authors may apply different quality criteria (with methods that can be improved in the future) if the original raw data is available.

I think that it could be interesting to justify how the sampling frequency was selected for each sensor: it was based mostly for operational limitations? or it was designed for the overall goals of the project?

---

## Referee Comment (RC2) · Anonymous Referee #2 · 31 Aug 2018

[referee-annotated manuscript omitted]

---

## Author Comment (AC1) · 22 Sep 2018

**Authors comment on* "Environmental conditions of a salt-marsh biodiversity experiment on the island of Spiekeroog (Germany)"**

First of all, we thank the two reviewers for their time and due diligence in reading and commenting on our manuscript. In the following we will reply to all referee comments (RC1 and RC2) with the reply in blue and changes in the manuscript in green.

**Review RC1, by Jaume Piera.**

In this contribution the authors report the abiotic parameters observed from 23 sensors installed around the facilities installed in the framework of the project BEFmate.

My major concern is about the data quality procedures. According to the method described by the authors, quality control was performed by a) erasing negative readings and data covering maintenance activities, b) visual inspection of the overall dataset and c) removal of outliers, defined as data exhibiting changes of more than two standard deviations within one time step.

Although the authors mention that data quality assurance and quality measures will be further developed, this primary quality control has some potential problems: "Visual inspection of the overall dataset" is always a fuzzy concept that provably change the criteria among different sensors. Furthermore, it is not reproducible.

At present there are several publications and guidelines to apply widely accepted criteria for data quality control of oceanographic data. See for example the guideline from SeaDataNet: https://www.seadatanet.org/Standards/Data-Quality-Control

Authors response: Thank you for highlighting this relevant aspect. The manual data inspection is a curation process that was used a) for cutting out time periods of sensor services and b) for time periods where sensors fell dry, due to their deployment in a tidal influenced area. The "visual inspection" mentioned in the manuscript was referring to these steps and there were no further "human decisions" involved. We erased the mentioning of "visual inspections" from the text, to avoid misunderstandings. The removal of outliers followed a clear concept, described in the manuscript. Raw data and Matlab routines are available on request. We also added that information to the manuscript.

Manuscript changes:

Section 2: …"b) visual inspection"… was removed from the manuscript, where mentioned.

In subsection 2.3 we added: "Raw data and Matlab routines applied in the curation process are available on request."

I suggest also to keep the original data and provide quality flags. It could be recommendable also to include a script to perform the quality actions to apply to the data (delete, interpolate, ...) based on the quality flag information. Using this approach, other authors may apply different quality criteria (with methods that can be improved in the future) if the original raw data is available.

Authors response: Thank you for that suggestion. The original data and Matlab routines used for data curation are archived at UOL servers and available on request. The quality-controlled data is available on Pangaea. We followed the concept of Pangaea database and ESSD that quality controlled and corrected data should be available to the scientific community. The reason behind this is, that raw data, even if flagged, might lead to false scientific interpretations, when downloaded by unexperienced users that do not understand and therefore ignore the flagging terminology. We added information to the manuscript, to clarify the availability of raw data and Matlab routines.

Manuscript changes:

In subsection 2.3 we added: "Raw data and MATLAB routines applied in the curation process are available on request."

I think that it could be interesting to justify how the sampling frequency was selected for each sensor: it was based mostly for operational limitations? or it was designed for the overall goals of the project?

Authors response: Thank you for raising that question. The sampling frequency was indeed a mixture of system limitations and the overall goal to measure tidal changes (the dominant tide is M2 with 12.4 h tidal period, therefore at minimum 48 samples per tide were assumed) over a long period between services. Service frequency in a UNESCO World Heritage Site is limited to reduce impacts on the environment. Therefore our access was limited to maximum one service per month. To avoid data loss if a service would be not possible based on e.g. weather conditions or breeding season, we designed the systems to be operable for at minimum two months. In all cases we could realize and exceed the required sampling (typ. 10 min sampling interval). We added a paragraph to the manuscript.

Manuscript changes:

In subsection 2.2.2 we added: "Sampling interval was chosen to resolve the tidal changes expected (dominant M2 lunar tide with 12.4 h period) while at the same time providing a minimum deployment time of two months between services, to account for the environmental protection regulations in this area. The same sampling interval was applied to the other sensors in this setup, except for the weather station."

**Review RC2, anonymous.**

The authors present a detailed explanation of an intertidal monitoring program from northern Germany from September 2014 to April 2017. The topic of biodiversity monitoring is introduced well and its importance in light of a changing climate. They provide a thorough background on their design set up (also with reference to prior work) and detail their monitoring equipment with information regarding their parameters and data acquisition. This is an extensive section that forms the basis of the paper and majority of the text. However, the authors do provide results from the various equipment recordings with accompanying figures and tables. The authors conclude with recommendations for further work in the region.

Authors response: Thank you for your positive feedback.

Minor improvements to the manuscript and figures could be made through comments uploaded in the associated pdf. The results section has no real discussion of their significance despite labelled "results and discussion'. Perhaps a discussion is the not the basis of the paper or journal though.

Authors response: Thank you for the detailed work on minor issues, that we will comment in the following and were adopted in the manuscript. Concerning the section "results and discussions" you are correct, that the term "discussions" is misleading if seen from the classical point of view of a scientific article. It is, as far as ESSD manuscript preparation guidelines explain it, not necessarily required. Here we used it as a reference to "discuss the data and results with respect to validity and potential limitations of the observational setup" as stated in the introduction (e.g. in section 4.2 or 4.4). The in depth scientific discussion is subject of related publications such as Balke et al. 2017.

Minor changes in the manuscript, according to the reviewers suggestions in the PDF:

- "salt marsh" without "-" throughout the manuscript

- DOIs in abstract is common practice in ESSD (not changed)

- Abstract extended with a final sentence and concluding statements: "A data availability of 80% for 17 out of 23 sensors was achieved. Results showed the influence of seasonal and tidal dynamics on the experimental islands. Nearby salt marsh plots were less influenced by these conditions and exhibited differences e.g. in temperature dynamics. Thus a consistent, multi-parameter, long-term data set is available as a basis for further biodiversity and ecosystem functioning studies."

Introduction:

- Updated reference: Rockström et al. (2009) Nature
- "As a consequence…"
- "intertidal" instead of "near-shore"
- References for salt marsh studies with respect to sea level rise added: Kirwan & Megonigal 2013; Balke et al. 2016

- "Based on this backdrop the project BEFmate "Biodiversity - ecosystem functioning across marine and terrestrial ecosystems" was conducted between Mar 2014 - Dec 2017 aiming to quantify …"
- consistent use of "experimental salt marsh islands" or "experimental islands" throughout the manuscript
- Sentence introducing the experimental islands and salt marsh plots was simplified, to not reflect methods (removal of technical details)

Materials and Methods:

- We kept "Research Area" in this section, since it does not contain sufficient information for a separate section.
- Figure 1: improved with labels "North Sea" and "German Bight". We did not provide a further North-West-European map since locations and landmarks are sufficiently provided. Country labels "Denmark" and "The Netherlands" added.
- …"east and west"… caps checked throughout manuscript
- "NHN (normal height null)"
- Figure 2: Labels added. Inserted panel magnified
- ""
- Double abbreviations erased throughout manuscript
- Figure 3: Caption extended to explain the patterns of the experimental island displayed "Experimental islands outer hull (galvanized steel) is patterned in the upper area to allow for a lateral water exchange. Lower area pattern is only meant to reduce weight and ease construction."

Results

- "wind roses" changed to "wind diagrams"
- "was observed" instead of "could be observed"
- ""
- Dates mentioned with 01st October or alike changed to 1st October
- Section 4.4.3 comma added to clarify meaning
- Figure 7 and 10: Colored lines broadened
- Figure 13: Legend enlarged to highlight color selection
- Table 9: Was not properly displayed in the current reproduction. Was reformatted.

Conclusions:

- "utilizing "
- ""

[revised manuscript text omitted]

All authors contributed to proof reading of the manuscript.

**Competing interests**

The authors declare that they have no conflict of interest.

**Acknowledgements**

The authors are very grateful to Kai Schwalfenberg and Claudia Thölen for their tremendous support in the field and laboratory. Sincere thanks to Daniela Voß, Ursel Gerken, Kathrin Dietrich, Nick Rüssmeier, Rohan Henkel, Jule Beßler, Franziska Wöhrmann and Hauke Haake for technical, laboratory or fieldwork support. Special thanks to Helmo Nicolai and Gerrit Behrens for their technical and logistical support. The support and cooperation with Nationalparkverwaltung Niedersächsisches Wattenmeer and the Umweltzentrum Wittbülten is acknowledged as well as the support of Rainer Sieger at PANGAEA. Thanks to the BEFmate colleagues and all other helping hands in the field during sampling campaigns, especially Regine Redelstein. The BEFmate project (Biodiversity - Ecosystem Functioning across marine and terrestrial ecosystems) was funded by the Ministry for Science and Culture of Lower Saxony, Germany under project number ZN2930. The feedback from two reviewers is gratefully acknowledged.

**References**

Balke, T., Stock, M., Jensen, K., Bouma, T.J., Kleyer, M.: A global analysis of the seaward salt marsh extent: The importance of tidal range, Water Resour. Res., 52, doi:10.1002/2015WR018318, 2016.

[revised manuscript text omitted]

---

## Editor Comment (EC1) · G. M. R. Manzella (Editor) · 24 Sep 2018

A further improvement is necessary to take into account the comment of Referee 1. It is suggested to authors to add some concepts on QA QC in the conclusions. The last sentence (Finally, data quality assurance and quality measures should be further developed to reduce the workload of manual data curation while improving data availability in near-real time) is not adding anything to the manuscript. The QA in a such experiment is requiring a lot of efforts. QA is essentially done before and during the observational efforts. A synthesis in the conclusion would help. QC is done when data are acquired. Near-real time and delayed mode QC require different efforts. The

authors are invited to better explain the methodologies they think to apply in future.

**[ESSDD](ESSDD)**